# FEDMEKI: A Benchmark for Scaling Medical Foundation Models via Federated Knowledge Injection

**Jiaqi Wang**[1*]    **Xiaochen Wang**[1*]    **Lingjuan Lyu**[2]    **Jinghui Chen**[1]    **Fenglong Ma**[1]

[1]Pennsylvania State University, [2]Sony AI

{jqwang, xcwang, jzc5917, fenglong}@psu.edu, lingjuan.lv@sony.com

https://github.com/psudslab/FEDMEKI

## Abstract

This study introduces the Federated Medical Knowledge Injection (FEDMEKI) platform, a new benchmark designed to address the unique challenges of integrating medical knowledge into foundation models under privacy constraints. By leveraging a cross-silo federated learning approach, FEDMEKI circumvents the issues associated with centralized data collection, which is often prohibited under health regulations like the Health Insurance Portability and Accountability Act (HIPAA) in the USA. The platform is meticulously designed to handle multi-site, multi-modal, and multi-task medical data, which includes 7 medical modalities, including images, signals, texts, laboratory test results, vital signs, input variables, and output variables. The curated dataset to validate FEDMEKI covers 8 medical tasks, including 6 classification tasks (lung opacity detection, COVID-19 detection, electrocardiogram (ECG) abnormal detection, mortality prediction, sepsis prediction, and enlarged cardiomediastinum detection) and 2 generation tasks (medical visual question answering (MedVQA) and ECG noise clarification). This comprehensive dataset is partitioned across several clients to facilitate the decentralized training process under 16 benchmark approaches. FEDMEKI not only preserves data privacy but also enhances the capability of medical foundation models by allowing them to learn from a broader spectrum of medical knowledge without direct data exposure, thereby setting a new benchmark in the application of foundation models within the healthcare sector.

## 1    Introduction

Foundation models have revolutionized various domains by demonstrating powerful capabilities in handling different modalities and tasks. Models such as GPT-3 [1] and LLaMA [2] have shown exceptional performance across a wide range of applications, including natural language processing, image classification, and multimodal reasoning. The primary reason for their success is their exposure to vast amounts of training data, enabling them to acquire a deep understanding of diverse domains. Leveraging this extensive data allows foundation models to generalize effectively and perform well across various tasks, making them invaluable in fields like healthcare, finance, and education. Furthermore, the scale of their training enables these models to capture nuanced relationships within data, enhancing their ability to perform high-level reasoning and decision-making. Consequently, foundation models serve as robust baselines and starting points for more specialized AI applications, fostering innovation and accelerating advancements across numerous domains.

In the medical domain, there have been attempts to develop medical foundation models that replicate the success seen in general domains [3, 4, 5]. However, the limited availability of public medical

---

*The first two authors contributed equally to this work.

38th Conference on Neural Information Processing Systems (NeurIPS 2024) Track on Datasets and Benchmarks.

data restricts the ability to train medical foundation models from scratch. To address this challenge, researchers have proposed fine-tuning general foundation models with medical data to customize medical foundation models. For instance, PMC-LLaMA [6] fine-tunes LLaMA with 4.8 million biomedical academic papers and 30,000 medical books. Similarly, LLaVA-Med [7] fine-tunes LLaVA [8] with biomedical image-text pairs extracted from PMC-15M [9]. Although existing medical foundation models have achieved superior performance on various domain-specific tasks, their scalability remains limited due to the current fine-tuning methods.

As previously discussed, most medical foundation models require fine-tuning existing general domain foundation models in a centralized training manner. However, due to the sensitivity and privacy issues of medical data, such centralized fine-tuning is unrealistic in real-world healthcare settings. Health regulations, such as the Health Insurance Portability and Accountability Act (HIPAA) in the USA, prohibit the collection and central storage of patient data for model training. In practice, medical data are stored at individual health institutions or hospitals and cannot typically be shared with others. Therefore, a more practical and realistic solution is to collaboratively inject medical knowledge learned from private client data into foundation models in a federated manner.

**A New Task.** To achieve this goal, we introduce a new task to scale existing medical foundation models, named **Fed**erated **Me**dical **K**nowledge **I**njection into foundation models (FEDMEKI). In this task, each client stores a set of private multi-modal, multi-task medical datasets, while the server hosts a medical foundation model. The objective is to inject client medical knowledge into the foundation model without sharing their private data. This new task presents several unique challenges compared to existing medical foundation model fine-tuning methods.

*C1 – Data Fine-tuning vs. Parameter Adaptation.* This new task prohibits the sharing of private data among clients. To extract medical knowledge from these clients, a straightforward solution is to treat the learned client model parameters as a new format of medical knowledge, which will be uploaded to the server for knowledge injection. However, the foundation model deployed on the server has different network structures from the client models, making it impossible to perform averaging operations like FedAvg [10]. The challenge here is to adapt client model parameters to the foundation model.

*C2 – Task-specific Fine-tuning vs. Scalable Fine-tuning.* Existing medical foundation models can only handle task-specific downstream tasks. For instance, LLaVA-Med is fine-tuned for medical vision question answering (VQA) tasks, including VQA-RAD [11], SLAKE [12], and PathVQA [13]. Similarly, PMC-LLaMA can only handle tasks that use text inputs, including PubMedQA [14], MedMCQA [15], and USMLE [16]. In addition to medical images and text, complex medical data include other commonly used modalities, such as medical signals and lab results, which existing medical foundation models often miss. Therefore, this new task is crucial for enabling the simultaneous fine-tuning of medical foundation models with diverse modalities.

**A Comprehensive Medical Dataset.** To address the aforementioned challenges and benchmark this new task, we first curated a new multi-site, multi-modal, multi-task dataset. This dataset covers **eight diverse medical tasks**: lung opacity detection [17], COVID-19 detection [18], ECG abnormal detection [19], mortality prediction [20], sepsis prediction [20], enlarged cardiomediastinum detection [21], MedVQA [11], and signal noise clarification [22]. These tasks span **seven medical modalities**: medical images, medical texts, medical signals, laboratory test results, vital signs, input variables, and output variables, extracted from **seven publicly available datasets** (RSNA [17], COVQU [18], PTB-XL [19], MIMIC-III [23], CheXpert [21], VQA-RAD [11], and ECG-QA [22]). We divided the tasks in our dataset into training tasks and validation tasks. The training tasks aim to inject modality-level knowledge into medical foundation models, while the validation tasks evaluate the ability of zero-shot inference for the knowledge-injected medical foundation models. The data is distributed to several clients, following a cross-silo federated learning setting similar to FLamby [24], due to the typically small size of medical datasets.

**A Novel Federated Knowledge Injection Platform.** We have developed a new FEDMEKI platform to address this new task with the curated dataset, as shown in Figure 1. Specifically, the platform is equipped with the functionalities of multi-modal multi-task data preprocessing, multi-site data partition, multi-modal multi-task client training, and medical foundation model federated scaling. Besides, it implements 16 methods as benchmarks to evaluate the platform, including traditional federated learning, federated learning with fine-tuning, and federated learning with foundation model scaling. To sum up, the contributions of this work are fourfold:

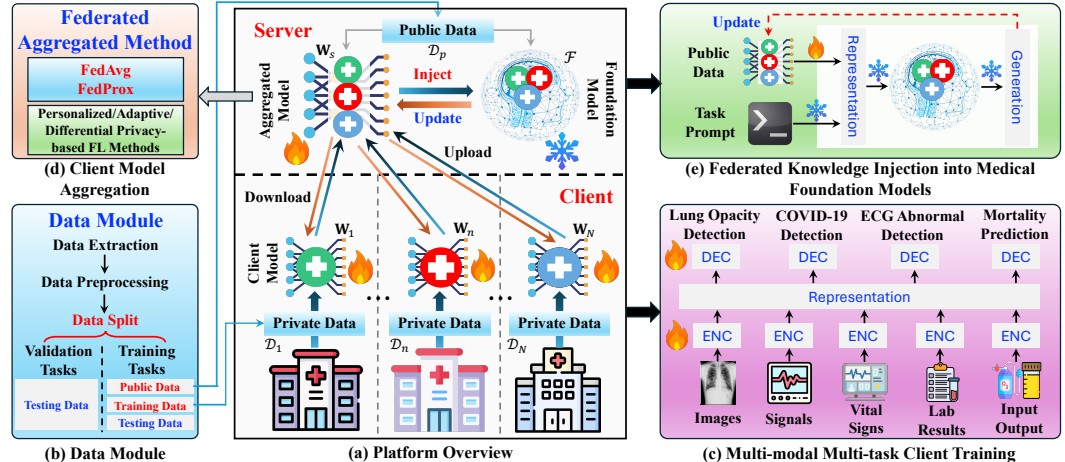

Figure 1: Overview of our proposed FEDMEKI platform.

- We investigate an important and practical task in the medical domain, aiming to inject medical knowledge into medical foundation models in a cross-silo federated manner, thereby scaling the capability of medical foundation models while ensuring privacy.
- We curate a new dataset from seven publicly available data sources, which covers eight diverse medical tasks (single-modal and multi-modal classification and generation tasks) with seven medical modalities.
- We build an open-source federated medical knowledge injection platform FEDMEKI for benchmarking this new task with the curated dataset. The FEDMEKI platform can be easily scaled with new medical tasks and integrates different federated learning algorithms.
- We implement 16 different approaches as benchmark baselines to validate the FEDMEKI platform in two scenarios: four training task evaluations to evaluate its task-specific capabilities and four validation task evaluations to assess its ability for zero-shot inference.

## 2 Related Work

**Federated Learning with Medical Data.** Medical data containing highly sensitive patient information is rigorously protected by various regulations and laws, making centralized access and processing impractical for machine learning model training. Federated learning [10, 25, 26, 27], a distributed paradigm, enables participants to train machine learning models without exchanging data. This approach has been extensively applied in medical tasks using different types of medical data, such as electronic health records (EHRs) [28, 29, 30, 31] and medical imaging [32, 33, 34]. There are a range of applications of federated learning in healthcare, encompassing disease prediction [35, 36, 37, 38], medical image classification [39, 40, 41], and segmentation [42, 43]. Additionally, several surveys have reviewed related advancements [44, 45, 46, 38]. To date, only one benchmark [24] has investigated the application of federated learning specifically to medical data. Notably, **no research** has yet explored the scalability of medical foundation models within a federated framework.

**Medical Foundation Models.** Foundation models, characterized by their extensive parameters and vast training datasets, have demonstrated remarkable capabilities across various domains [2, 47, 48, 49, 50]. In the realm of healthcare [51, 52], these models are increasingly prevalent. Thirunavukarasu et al. (2023) [53] discuss the potential of large language models (LLMs) in clinical settings, highlighting their effectiveness in healthcare applications. Moor et al. (2023) [3] introduce the concept of a generalist medical AI, designed to handle diverse tasks using multimodal medical data. Additionally, specialized medical foundation models have been developed for targeted applications such as disease detection using retinal images [5], cancer imaging biomarker identification [54], echocardiogram interpretation [55], medical image segmentation [56], and precision oncology [57]. Despite these advancements, there remains **a gap in research** concerning the development of datasets and benchmarks that enable medical foundation models to integrate and leverage medical knowledge from distributed data sources.

**Federated Fine-tuning with Foundation Models.** To achieve better performance in specific tasks, fine-tuning foundation models (FMs) with task-specific data is essential. FL facilitates this fine-tuning process by allowing the use of locally stored data through distributed computational resources [58]. Existing related research can be categorized into full tuning [59, 60], partial tuning [61, 62, 63], and parameter-efficient fine-tuning (PEFT) [64, 65]. In [64], each client has a foundation model and exchanges the adapters with the server in each communication round. The server conducts the basic FedAvg on the adapter and sends it back to the clients. Similarly, FedPETuning [65] provides a PEFT approach on pre-trained language models via sharing part of the client models in FL. The aforementioned studies typically require clients to possess FMs, with the aim of mutual benefits. In contrast, our approach places the medical FM on the server side, representing a more practical setting. Moreover, our objective is to enable clients to collaboratively contribute to **scaling the capability** of the medical FM models without accessing local data.

## 3 The FEDMEKI Platform

As shown in Figure 1(a), the designed FEDMEKI platform consists of several clients $\{C_1, \cdots, C_n\}$ and a server $S$. Each client $C_n$ trains a specific model $\mathbf{W}_n$ using private data $\mathcal{D}_n$, which can be treated as the knowledge representation of the client. The trained client models $\{\mathbf{W}_1, \cdots, \mathbf{W}_N\}$ will be uploaded to the server. After receiving the client models, the server will inject the aggregated medical knowledge representation by $\mathbf{W}_s$ into the medical foundation model $\mathcal{F}$ using the public data $\mathcal{D}_p$. The updated global model $\mathbf{W}_s$ will be distributed to each client again for the learning of the next communication round until convergence.

### 3.1 Client Deployment

The goal of FEDMEKI is to inject medical knowledge learned from private multi-modal multi-task data $\mathcal{D}_n$ into the foundation model $\mathcal{F}$. We deploy a basic client model $\mathbf{W}_n$ to handle the multi-modal multi-task data to achieve this goal.

**Modality-specific Encoders.** Although we have five training tasks for each client, some tasks share the same modality. For example, both ECG abnormal detection [19] and ECGQA [22] tasks have the signal ECG modality. To avoid the redundancy of modality encoders and learn shared features across tasks, we propose to deploy modality-specific encoders. The details of these encoders are shown in Appendix Section N. Let $(\mathbf{x}_n^i, \mathbf{y}_n^i) \in \mathcal{D}_n$ denote a training sample. Only the task-associated encoders will generate outputs, and the output of an encoder is denoted as $\text{ENC}_n^m(\mathbf{x}_n^i)$ ($m \in [1, M]$), where $M$ is the number of unique modalities. We finally obtain the task-specific representation of each data sample $\mathbf{r}_n^i$ by concatenating outputs from task-associated encoders.

**Task-specific Decoders.** Each task has a unique decoder $\text{DEC}_n^t(\mathbf{r}_n^i)$ to generate the outcome and we use cross-entropy as the loss. The details of each task-specific decoder are shown in Appendix of Section N.

**Federated Optimization.** The ground truth $\mathbf{y}_n^i$ will be used to optimize the client model $\mathbf{W}_n$ with the cross-entropy loss for all training tasks. Since there are several ways to conduct federated learning, we use FedAvg [10] and FedProx [66] as examples to demonstrate how FEDMEKI works in this study.

• **FedAvg** [10] aims to collaboratively train each client separately and upload their model parameters $\{\mathbf{W}_1, \cdots, \mathbf{W}_N\}$ directly to the server.

• **FedProx** [66] is developed based on FedAvg but added an $L_2$ regularization term on each local loss function as follows

$$\min_{\mathbf{W}_n} \mathcal{J}_n(\mathbf{W}_n; \mathbf{W}_s) = \mathcal{L}_n(\mathbf{W}_n) + \frac{\lambda}{2}||\mathbf{W}_n - \mathbf{W}_s||^2, \tag{1}$$

where $\mathbf{W}_s$ is the global model, $\mathcal{L}_n(\cdot)$ is the client loss function, and $\lambda$ is a hyperparameter. The learned client parameters $\{\mathbf{W}_1, \cdots, \mathbf{W}_N\}$ will be uploaded to the server. Since the designed FEDMEKI platform is general, we can use any FedAvg-style approaches, including *personalized FL* methods [67, 68], *differential privacy-based FL* methods [69, 70], and *adaptive FL* methods [71, 72, 73].

## 3.2 Server Deployment

We deploy a model aggregator on the server to aggregate client models $\{\mathbf{W}_1, \cdots, \mathbf{W}_N\}$ and a LLaVA-style module to inject medical knowledge with the help of public data.

**Client Model Aggregation.** We still follow FedAvg-style approaches to obtain the aggregated global model $\mathbf{W}_s$ using the averaging of all client models, i.e., $\mathbf{W}_s = \frac{1}{N} \sum_{n=1}^{N} \mathbf{W}_n$.

**Scaling Medical Foundation Model $\mathcal{F}$.** We deploy a medical foundation model $\mathcal{F}$ on the server. Note that it can be *any of the existing medical foundation models*, such as MedVInT [74] and ChatDoctor [75]. The current platform uses MMedLM-2[2] [76] as $\mathcal{F}$, which is a pretrained language model for medicine and achieves state-of-the-art performance on several tasks. MMedLM-2 can only take text as the input. Our goal is to enable $\mathcal{F}$ to work on tasks with other modalities.

To this end, we follow the LLaVA's fine-tuning style to generate the representation $\mathbf{r}_p^j$ of a public data sample $(\mathbf{x}_p^j, \mathbf{y}_p^j) \in \mathbf{D}_p$ using the encoder of $\mathbf{W}_s$ first. We then align $\mathbf{r}_p^j$ with the task prompt representation $\mathbf{t}_k$ using a linear layer, i.e., $\mathbf{h}_p^j = \mathrm{MLP}(\mathbf{r}_p^j)$, where $\mathbf{t}_k = \mathrm{EMB}_{\mathcal{F}}(\mathcal{T}_k)$, $\mathrm{EMB}_{\mathcal{F}}(\cdot)$ denotes the embedding layer of $\mathcal{F}$, and $\mathcal{T}_k$ is the $k$-th task's prompt. The concatenation of $\mathbf{h}_p^j$ and $\mathbf{t}_k$ is subsequently fed into $\mathcal{F}$ to generate the output $\hat{\mathbf{y}}_p^j$. Finally, the parameters are optimized by the ground truth $\mathbf{y}_p^j$. Note that all parameters of $\mathcal{F}$ are fixed during the optimization, and only the encoder of $\mathbf{W}_s$ will be updated. The updated $\mathbf{W}_s$ is then sent to all clients again for updates in the next communication round until FEDMEKI converges.

## 4 The FEDMEKI Dataset Suite

Since we propose a new research task, **no** existing datasets are suitable for evaluation. We curated a new dataset from publicly available medical sources to address this, comprising two types of tasks: training and validation. The **training tasks** are used to scale the medical foundation model and to evaluate its task-specific capabilities. The **validation tasks** are *independent* of the training tasks and are used to assess the ability of the scaled medical foundation model in zero-shot inference.

### 4.1 Training Tasks

To inject medical knowledge into the foundation model $\mathcal{F}$, as shown in Section 3.1, we need to train tasks to cover as many medical modalities as possible. In this benchmark, we choose 4 commonly used classification tasks covering 6 medical modalities. Note that we do not use any tasks with the text modality since the medical foundation model $\mathcal{F}$ has the superior capability to handle texts.

(1) Lung Opacity Detection [17] is an unimodal classification task aiming at predicting lung opacity from chest X-ray **images**. The data are provided by the RSNA Pneumonia Detection Challenge 2018 [17]. Medical practitioners at the Society for Thoracic Radiology and MD.ai provide the annotations, i.e., ground truth labels. The original medical images are found in the chest X-ray database [77]. The data details are in Appendix Section F.

(2) COVID-19 Detection requires the model to determine whether an X-ray **image** indicates COVID-19 symptoms, testing the model's understanding of medical images. We utilize the COVQU dataset [18] for this task. The details of this task can be found in Appendix Section G.

(3) ECG Abnormal Detection aims to determine whether an electrocardiogram (ECG) **signal** exhibits abnormal patterns or not. This is an unimodal binary classification task, where the data are sourced from an existing ECG database [19], consisting of 12-lead ECGs of 10-second length. The data details are in Appendix Section H.

(4) Mortality Prediction involves using various data points and a classification or predictive model to estimate the likelihood of a patient's survival or death during their stay in the Intensive Care Unit (ICU). We extract the data from MIMIC-III using the ICU-oriented preprocessing pipeline [78]. Following [20], we extract 48 dynamic features, including **vital signs** (7 variables) and **laboratory tests** (39 variables), with **two more variables** that measure input (*fraction of inspired oxygen*) and output (*urine*). The data details are in Appendix Section I.

---

[2] https://huggingface.co/Henrychur/MMedLM2

Table 1: Details of data split, where we deploy 5 clients on the FEDMEKI platform.

| Type | Task | Total Samples | Total Training (5 Clients) | Public Data (Server) | Development (Server) | Testing (Server) |
|---|---|---|---|---|---|---|
| Training Tasks | Lung Opacity Detection | 18,406 | 12,880 | 1,849 | 1,841 | 1,836 |
| | COVID-19 Detection | 13,808 | 9,665 | 1,380 | 1,380 | 1,383 |
| | ECG Abnormal Detection | 21,797 | 15,259 | 2,179 | 2,180 | 2,179 |
| | Mortality Prediction | 38,129 | 26,690 | 3,812 | 3,812 | 3,813 |
| Validation Tasks | Enlarged Cardiomediastinum Detection | 234 | ✗ | ✗ | ✗ | 234 |
| | Sepsis Prediction | 1,000 | ✗ | ✗ | ✗ | 1,000 |
| | MedVQA | 1,000 | ✗ | ✗ | ✗ | 1,000 |
| | Signal Noise Clarification | 1,000 | ✗ | ✗ | ✗ | 1,000 |

## 4.2 Validation Tasks

Using the training tasks, we can inject various medical knowledge into the foundation model $\mathcal{F}$ by inserting an aggregated encoder learned from federated clients into $\mathcal{F}$. We use four new tasks to evaluate the generalization ability of the federated scaled $\mathcal{F}$ learned by the FEDMEKI platform with 2 classification tasks and 2 generation tasks.

(5) Enlarged Cardiomediastinum Detection [21] aims to determine the likelihood of an enlarged cardiomediastinum using medical **images** from clinical assessments. This task evaluates the model's ability to interpret radiographic data. Further details of this task can be found in Appendix Section J.

(6) Sepsis Prediction aims to predict the probability of sepsis occurring during ICU stays, examining the model's ability to comprehend diverse **clinical features**, which are the same as those extracted for the mortality prediction task from the MIMIC-III database using the preprocessing pipeline [20]. The details of this task can be found in Appendix Section K.

(7) Medical Visual Question Answering (MedVQA) aims to use both **visual images** and **textual questions** as inputs to generate the answers. This task tests the model's ability to align text and image modalities in the medical domain. We use the VQA-RAD dataset in this work [11]. The details of this task can be found in Appendix Section L.

(8) Signal Noise Clarification is another generative task that focuses on accurately describing noise in ECG **signals** with the corresponding **textual questions**, where the data are extracted from an existing ECG question answering dataset [22]. The signals are in 12 channels, lasting 10 seconds, similar to the ECG Abnormal Detection task. The data details are in Appendix Section M.

## 4.3 Data Partition

The **training tasks** have two roles. The first role is to inject the medical knowledge in the training tasks into the foundation model $\mathcal{F}$. The second one is to evaluate the performance of these training tasks on the scaled $\mathcal{F}$. Thus, for each training task, we divide the data into four parts in a ratio of 7:1:1:1, where 70% data $\mathcal{D}_{tr}^{tra}$ are the real training data that will be evenly distributed to $N$ clients, 10% data as the public data $\mathcal{D}_{p}^{tra}$ that will be put on the server, another 10% data as the development data $\mathcal{D}_{d}^{tra}$ that are preserved on the server to guide the model training, and the remaining 10% data $\mathcal{D}_{te}^{tra}$ as the testing data for training tasks. The **validation tasks** aim to evaluate the capability of zero-shot inference. For validation tasks with numerous samples in the test set, we randomly choose several data samples $\mathcal{D}_{te}^{val}$ for the testing. Details of these datasets' split are available in Table 1.

# 5 Benchmark

## 5.1 Approaches & Evaluation Metrics

We use the following approaches as benchmarks for the evaluation of **training tasks**, which will be evaluated with the training data of the training tasks, i.e., $\mathcal{D}_{tr}^{tra}$. Our evaluation focuses on two scenarios: single-task and multi-task evaluations. Note that the original medical foundation model MMedLM-2, which can only input text data, cannot work on all these tasks.

Table 2: Benchmark performance of single-task evaluation for training tasks.

| Task | Metric | MMedLM-2 | FedAvg | | | | FedProx | | | |
|---|---|---|---|---|---|---|---|---|---|---|
| | | | $\text{FedAvg}_s$ | $\text{FedAvg}_s^+$ | $\text{FedAvg}_s^*$ | $\text{FedAvg}_s^{\mathcal{F}}$ | $\text{FedProx}_s$ | $\text{FedProx}_s^+$ | $\text{FedProx}_s^*$ | $\text{FedProx}_s^{\mathcal{F}}$ |
| Lung Opacity Detection | Accuracy | ✗ | 95.86 | 94.44 | 96.02 | 89.42 | 95.70 | 96.08 | 95.70 | 91.23 |
| | Precision | ✗ | 97.40 | 93.81 | 96.70 | 84.69 | 97.49 | 97.11 | 95.23 | 87.76 |
| | Recall | ✗ | 94.01 | 95.58 | 95.58 | 97.16 | 94.11 | 95.27 | 96.53 | 96.52 |
| | F1 | ✗ | 95.31 | 94.69 | 96.14 | 90.50 | 95.77 | 96.18 | 95.87 | 91.93 |
| COVID-19 Detection | Accuracy | ✗ | 99.35 | 99.48 | 99.28 | 92.34 | 99.13 | 99.42 | 99.13 | 84.16 |
| | Precision | ✗ | 99.71 | 99.70 | 100.00 | 93.59 | 99.71 | 99.42 | 99.71 | 77.27 |
| | Recall | ✗ | 97.72 | 94.30 | 97.15 | 74.92 | 96.87 | 98.29 | 96.87 | 53.27 |
| | F1 | ✗ | 98.71 | 96.93 | 98.55 | 79.15 | 98.21 | 98.85 | 98.27 | 63.07 |
| ECG Abnormal Detection | Accuracy | ✗ | 67.68 | 66.83 | 57.86 | 43.15 | 79.41 | 80.51 | 57.77 | 45.25 |
| | Precision | ✗ | 69.13 | 80.65 | 89.56 | 56.97 | 89.04 | 89.06 | 87.34 | 60.85 |
| | Recall | ✗ | 80.78 | 56.24 | 31.61 | 11.22 | 73.88 | 76.00 | 32.47 | 17.80 |
| | F1 | ✗ | 74.50 | 66.27 | 46.72 | 18.74 | 80.75 | 82.01 | 47.34 | 27.55 |
| Mortality Prediction | Accuracy | ✗ | 91.98 | 91.66 | 91.61 | 84.11 | 91.98 | 90.12 | 91.61 | 82.41 |
| | Precision | ✗ | 70.00 | 52.86 | 58.33 | 16.35 | 71.05 | 36.45 | 58.33 | 13.87 |
| | Recall | ✗ | 8.70 | 11.42 | 2.17 | 21.43 | 8.39 | 22.98 | 2.17 | 16.64 |
| | F1 | ✗ | 15.47 | 18.88 | 4.19 | 18.55 | 15.00 | 28.19 | 4.19 | 15.13 |

**Eight Single-task Evaluation Benchmarks.** Single-task evaluation aims to validate the generalization ability of FEDMEKI on tasks with specific modalities. We use the following approaches as benchmark baselines: (1) Traditional Federated Learning (TFL). We use two representative federated learning models as benchmark baselines: FedAvg [10] and FedProx [66]. For each task, we use the corresponding task data to train an FL model $\textbf{FedAvg}_s$ or $\textbf{FedProx}_s$. We use the aggregated global model to evaluate the performance. (2) Federated Learning with Global Fine-tuning (FL+GF). Since the server stores a small set of public data $\mathcal{D}_p^{tra}$, the traditional models can conduct the fine-tuning using $\mathcal{D}_p^{tra}$ for the aggregated global models. These approaches are denoted as $\textbf{FedAvg}_s^+$ and $\textbf{FedProx}_s^+$. (3) Federated Learning with LLM Fine-tuning (FL+LLM). To further enhance the learning ability of traditional federated learning approaches, we allow them to fine-tune with the LLM. In particular, the encoder of each aggregated model will be used first to generate the representation of the public data. The representation is then concatenated with the representation of LLM to generate the output. We denote these LLM fine-tuning approaches as $\textbf{FedAvg}_s^{\mathcal{F}}$ and $\textbf{FedProx}_s^{\mathcal{F}}$. Besides, we can obtain the aggregated models from $\textbf{FedAvg}_s^{\mathcal{F}}$ and $\textbf{FedProx}_s^{\mathcal{F}}$ on the server as traditional FL approaches, denoted as $\textbf{FedAvg}_s^*$ and $\textbf{FedProx}_s^*$.

**Eight Multi-task Evaluation Benchmarks.** The final goal of the designed FEDMEKI platform is to evaluate the multi-site, multi-modal, multi-task medical knowledge injection. Since MMedLM-2 can only handle single modality inputs, we do not consider baselines of directly using MMedLM-2 in this evaluation. (1) TFL. We still employ FedAvg [10] and FedProx [66] but use a multi-modal multi-task encoder for each client model as described in Section 3.1. These two approaches are denoted as $\textbf{FedAvg}_m$ and $\textbf{FedProx}_m$. (2) FL+GF. We can also fine-tune the aggregated model on the server using the public data $\mathcal{D}_p^{tra}$ at each communication round. We use $\textbf{FedAvg}_m^+$ and $\textbf{FedProx}_m^+$ to denote the fine-tuned approaches. (3) FL+LLM. We use $\textbf{FedAvg}_m^{\mathcal{F}}$ and $\textbf{FedProx}_m^{\mathcal{F}}$ to denote the federated fine-tuned approaches, which are the full version of solutions deployed on the proposed FEDMEKI platform. Except for the fine-tuned medical foundation models, we can also obtain an aggregated global model, denoted as $\textbf{FedAvg}_m^*$ or $\textbf{FedProx}_m^*$, similar to traditional FL.

The details of all these 16 benchmark approaches can be found in Appendix Section N.

**Low-resource Evaluation Benchmarks.** We have four **validation tasks** with diverse modalities. Without federated scaling of the original medical foundation model, MMedLM cannot handle these three tasks. Thus, we use the scaled medical foundation models, including $\textbf{FedAvg}_m^{\mathcal{F}}$, and $\textbf{FedProx}_m^{\mathcal{F}}$ to evaluate the three validation tasks with zero-shot inference on $\mathcal{D}_{te}^{val}$.

**Evaluation Metrics.** We use accuracy, precision, recall, and F1 as the evaluation metrics for the classification tasks, and BLEU, ROUGE, and METEOR are used to evaluate the generation tasks. The higher, the better.

## 5.2 Benchmark Results

### 5.2.1 Evaluation Results of Training Tasks

**Single-task Benchmarks.** Table 2 shows the results of the single-task benchmarks. We can observe that the existing medical foundation model MMedLM-2 cannot handle these tasks. However, after

Table 3: Benchmark performance of multi-task evaluation for training tasks. Note that the performance of **ECG Abnormal Detection** and **Mortality Prediction** is the same as that shown in Table 2 since the modalities of these two tasks are non-overlapped with others.

| Task | Metric | MMedLM-2 | FedAvg | | | | FedProx | | | |
|---|---|---|---|---|---|---|---|---|---|---|
| | | | $\text{FedAvg}_m$ | $\text{FedAvg}_m^+$ | $\text{FedAvg}_m^*$ | $\text{FedAvg}_m^{\mathcal{F}}$ | $\text{FedProx}_m$ | $\text{FedProx}_m^+$ | $\text{FedProx}_m^*$ | $\text{FedProx}_m^{\mathcal{F}}$ |
| Lung Opacity Detection | Accuracy | ✗ | 95.42 | 94.23 | 94.88 | 95.48 | 94.77 | 96.24 | 96.41 | 93.13 |
| | Precision | ✗ | 99.66 | 93.51 | 93.68 | 98.22 | 99.54 | 97.22 | 97.63 | 93.74 |
| | Recall | ✗ | 91.48 | 95.48 | 96.64 | 92.95 | 90.33 | 95.48 | 95.37 | 92.95 |
| | F1 | ✗ | 95.39 | 94.48 | 95.13 | 95.51 | 94.71 | 96.34 | 96.49 | 93.35 |
| COVID-19 Detection | Accuracy | ✗ | 99.06 | 98.99 | 99.28 | 98.34 | 99.06 | 99.20 | 98.99 | 86.11 |
| | Precision | ✗ | 99.42 | 98.56 | 99.14 | 96.07 | 99.13 | 98.85 | 98.56 | 65.09 |
| | Recall | ✗ | 96.87 | 97.44 | 98.01 | 97.44 | 97.15 | 98.01 | 97.44 | 97.72 |
| | F1 | ✗ | 98.12 | 97.99 | 98.57 | 96.75 | 98.13 | 98.43 | 97.99 | 78.13 |

scaling it with private medical data on the designed FEDMEKI platform, the scaled models $\text{FedAvg}_s^{\mathcal{F}}$ and $\text{FedProx}_s^{\mathcal{F}}$ can work for these training tasks. These comparisons demonstrate that the FEDMEKI platform effectively achieves the goal of medical knowledge injection.

We can also observe that the federated scaled medical foundation models, $\text{FedAvg}_s^{\mathcal{F}}$ and $\text{FedProx}_s^{\mathcal{F}}$, still perform worse on the four training tasks than traditional federated learning approaches, $\text{FedAvg}_s$ and $\text{FedProx}_s$, and their scaled version $\text{FedAvg}_s^+$ and $\text{FedProx}_s^+$. This is reasonable since they are specifically designed for federated learning, and the aggregated global models do not contain any "noisy knowledge" injected by the medical foundation model MMedLM-2. However, comparing their performance is not the goal of this work. We aim to enable the medical foundation model to handle tasks with diverse medical modalities.

$\text{FedAvg}_s^*$ and $\text{FedProx}_s^*$ are the byproducts of $\text{FedAvg}_s^{\mathcal{F}}$ and $\text{FedProx}_s^{\mathcal{F}}$. Their performance is comparable to that of federated learning approaches on two image classification tasks but worse on the other two tasks. This may be because these two tasks are easier than ECG abnormal detection and mortality prediction tasks, and the medical foundation model can also be quickly adapted to these easy tasks.

**Multi-task Benchmarks.** Although we train multiple tasks with a designed multi-modal multi-task encoder, two of these tasks (ECG abnormal detection and mortality prediction) do not share overlapped modalities, leading to the same performance as single-task training as shown in Table 2. Thus, we do not list them in Table 3. We can observe that for the two image classification tasks, both foundation models, $\text{FedAvg}_m^{\mathcal{F}}$ and $\text{FedProx}_m^{\mathcal{F}}$, significantly improve their performance compared with single-task benchmarks, $\text{FedAvg}_s^{\mathcal{F}}$ and $\text{FedProx}_s^{\mathcal{F}}$. These results clearly demonstrate the importance and necessity of training multiple medical tasks together when injecting medical knowledge into foundation models.

### 5.2.2 Evaluation Results of Validation Tasks

**Low-resource Benchmarks.** A primary goal of training foundation models is to boost the performance of multiple downstream tasks, especially for zero-shot inference. To achieve this goal, we test the scaled medical foundation models in the previous experiment with four tasks. The enlarged cardiomediastinum detection task is similar to the lung opacity prediction task, as both take radiological images as input. Also, the sepsis prediction task is similar to the mortality prediction task in training, sharing the same feature space. However, the MedVQA and signal noise clarification tasks are new since they combine two modalities, which were not trained during the training. Thus, the two generation tasks are much harder than the two classification ones.

From the results shown in Table 4, we can observe that the knowledge-injected medical foundation models have the ability to deal with new tasks. Although the performance of the two generation-based tasks still has significant room for improvement, the designed platform at least can work for such tasks compared to the original medical foundation model MMedLM-2. Therefore, these results still demonstrate the utility of our benchmark for federated medical knowledge injection.

## 6 Discussion

**Summary of Key Findings.** In this study, we aimed to create a benchmark for federated medical knowledge injection into medical foundation models. To achieve this, we curated a comprehensive

Table 4: Zero-shot evaluation for validation tasks.

| Task (Modalities) | Metric | MMedLM-2 | FedAvg$_m^{\mathcal{F}}$ | FedProx$_m^{\mathcal{F}}$ |
|---|---|---|---|---|
| Enlarged Cardiomediastinum Detection (medical image) | Accuracy | ✗ | 58.54 | 57.26 |
| | Precision | ✗ | 53.33 | 52.57 |
| | Recall | ✗ | 88.07 | 84.40 |
| | F1 | ✗ | 66.04 | 64.78 |
| Sepsis Prediction (48 clinical features) | Accuracy | ✗ | 39.00 | 39.80 |
| | Precision | ✗ | 2.61 | 3.57 |
| | Recall | ✗ | 55.17 | 75.86 |
| | F1 | ✗ | 4.98 | 6.81 |
| MedVQA (medical image + text) | BLEU | ✗ | 1.20 | 1.20 |
| | ROUGE | ✗ | 2.43 | 3.42 |
| | METEOR | ✗ | 1.07 | 2.83 |
| Signal Noise Clarification (signal + text) | BLEU | ✗ | 0.06 | 0.04 |
| | ROUGE | ✗ | 0.29 | 0.23 |
| | METEOR | ✗ | 1.88 | 0.63 |

dataset for evaluation and implemented 16 benchmark baselines. Our enhanced foundation models demonstrated the capability to handle new tasks involving new medical modalities, showcasing the potential of this approach. However, the performance of these new foundation models was observed to be lower compared to traditional federated learning models.

**Implications of the Study.** Our findings have several important implications for the field of medical AI. Firstly, the ability of the enhanced foundation models to adapt to new medical modalities without the need for retraining from scratch highlights the potential for more efficient and scalable AI systems in healthcare. This capability can lead to significant time and resource savings, particularly in rapidly evolving medical fields. Secondly, federated learning ensures data privacy and security, which is paramount in handling sensitive medical data. The creation of a curated dataset and implementation of 16 benchmark baselines provide a robust framework for evaluating the effectiveness of federated medical knowledge injection, setting a standard for future research in this area.

**Limitations.** Our study has several limitations. The primary limitation is the observed performance trade-off when injecting medical knowledge into the foundation models. Moreover, the performance of zero-shot evaluation is still unsatisfactory. Additionally, the diversity and quality of the data available from multiple clients could impact the learning outcomes. Federated learning introduces challenges related to communication overhead and synchronization across clients, which might affect the overall efficiency and effectiveness of the learning process.

**Future Research Directions.** Future research should focus on optimizing the training algorithms to better handle the increased complexity introduced by the injection of medical knowledge. Exploring advanced federated learning techniques, such as personalized federated learning or federated transfer learning, could potentially enhance performance. Additionally, investigating more efficient communication protocols and strategies to manage data heterogeneity across clients would be beneficial. Expanding the study to include a wider variety of medical modalities and tasks could further validate the versatility and robustness of the proposed approach. Moreover, continually refining the curated dataset and updating the benchmark baselines will be crucial for ongoing evaluation and improvement.

# 7   Conclusion

Our study demonstrates the potential of injecting medical knowledge into foundation models within a federated learning framework. While there are challenges related to performance optimization, the enhanced adaptability and scalability of these models represent a promising direction for future medical AI research. By addressing the current limitations and exploring advanced learning techniques, we can further improve the efficacy and application of these innovative models in healthcare. Our curated dataset and benchmark baselines provide a solid foundation for continued research and development in this area.

**Acknowledgements** This work is partially supported by the National Science Foundation under Grant No. 2348541 and 2238275.

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

# Contents

## A Broader Impact

The Federated Medical Knowledge Injection (FMKI) platform introduces a transformative approach in healthcare AI, addressing critical issues of data privacy and accessibility by leveraging federated learning to inject medical knowledge into foundation models. This method not only complies with stringent health regulations, thereby protecting patient confidentiality, but also enhances the scalability and adaptability of medical foundation models. By enabling these models to utilize diverse, multi-modal medical data without direct data sharing, FMKI significantly broadens the potential applications of AI in healthcare, offering improved diagnostic accuracy and personalized treatment options. Furthermore, the platform facilitates equitable technology access, allowing institutions with varying resources to participate in and benefit from cutting-edge medical AI developments. This innovative approach not only promises to improve global healthcare outcomes but also sets new benchmarks in the ethical development and deployment of AI technologies in sensitive sectors.

## B Compute and Environment Configuration

All experiments are conducted on an NVIDIA A100 with CUDA version 12.0, running on a Ubuntu 20.04.6 LTS server. More details can be found in the GitHub repository.

## C Platform Repository

We have established a GitHub repository, available at `https://github.com/psudslab/FEDMEKI`. This repository includes resources for data processing, baselines, environmental setup, our proposed platform, and sample execution scripts. All the details have been documented at the ReadMe file[3]. We are committed to continuously updating this repository with additional modalities, datasets, and tasks.

## D Author Statement

As authors of this repository and article, we bear all responsibility in case of violation of rights and licenses. We have added a disclaimer on the repository to invite original dataset creators to open issues regarding any license-related matters.

---

[3]`https://github.com/psudslab/FEDMEKI/blob/main/README.md`

# E  Datasheet for Datasets

## E.1  Motivation

- **For what purpose was the dataset created?**
  This work investigates a novel yet practical task – scaling existing medical foundation models by injecting diverse medical knowledge with distributed private medical data. However, no available datasets are suitable for evaluation. Thus, we curated a new multi-site, multi-modal, and multi-task dataset, including five training tasks and three validation tasks and covering six commonly used medical modalities.

- **Who created the dataset (e.g., which team, research group) and on behalf of which entity (e.g., company, institution, organization)?**
  The authors of this paper.

- **Who funded the creation of the dataset? If there is an associated grant, please provide the name of the granter and the grant name and number.**
  This work is partially supported by the National Science Foundation under Grant No. 2238275, 2333790, 2348541, and the National Institutes of Health under Grant No. R01AG077016.

## E.2  Composition

- **What do the instances that comprise the dataset represent (e.g., documents, photos, people, countries)?**
  The FEDMEKI data suit contains medical images and the corresponding annotations for the pneumonia detection and COVID-19 detection tasks; ECG signals and the labels for the ECG abnormal detection task; 48 clinical features for the mortality prediction and sepsis prediction tasks; medical text (questions, candidate answers, document collections, ground truths) for the MedQA task; ECG signals, questions and answers for the signal noise clarification task; and medical images, questions, and answers for the MedVQA task.

- **How many instances are there in total (of each type, if appropriate)?**
  The Lung Opacity Detection task has 18,406 samples, the ECG Abnormal Detection task has 21,797 samples, and the Mortality Prediction task has 38,129 samples. The COVID-19 Detection task has 13,808 samples. The MedVQA, Signal Noise Clarification and Sepsis Prediction tasks each contain 1,000 samples. Additionally, the Enlarged Cardiomediastinum Detection task has 234 samples. Detailed information about the data can be found in Table 1.

- **Does the dataset contain all possible instances or is it a sample (not necessarily random) of instances from a larger set?**
  The ECG Abnormal Detection task includes all available samples from its corresponding database. The Lung Opacity Prediction, COVID-19 Detection, and Mortality Prediction tasks encompass all data samples with available binary labels, making them subsets of the original dataset. For validation tasks without a predefined test set or with an excessively large test set, we randomly selected 1,000 samples for testing. These tasks include MedVQA, Signal Noise Clarification, and Sepsis Prediction. For the Enlarged Cardiomediastinum Detection task, the original database provided a small test set of 234 samples, which we have retained.

- **What data does each instance consist of?**
  The Lung Opacity Prediction, COVID-19 Detection, and Enlarged Cardiomediastinum Detection tasks involve radiological images, while the ECG Abnormal Detection task involves 12-channel, 10-second ECG signals. The Mortality Prediction and Sepsis Prediction tasks cover temporal features involving vital signs, lab events, and input/output data. The Signal Noise Clarification task includes signal-text pairs, while the MedVQA task comprises image-text pairs.

- **Is there a label or target associated with each instance?**
  The answer (label) is provided for each instance.

- **Is any information missing from individual instances? If so, please provide a description, explaining why this information is missing (e.g., because it was unavailable). This does not include intentionally removed information, but might include, e.g., redacted**

**text.**
No.

- **Are relationships between individual instances made explicit (e.g., users' movie ratings, social network links)?**
  No.

- **Are there any errors, sources of noise, or redundancies in the dataset?**
  Questions are created by filling the slots in the templates with pre-defined values and records from the database. Thus, some questions can be grammatically incorrect but not critical (e.g., verb tense).

- **Is the dataset self-contained, or does it link to or otherwise rely on external resources (e.g., websites, tweets, other datasets)?**
  The proposed dataset depends on several open-source databases: RSNA [17], COVQU [18], PTB-XL [19], MIMIC-III [23], CheXpert [21], VQA-RAD [11], and ECG-QA [22].

- **Does the dataset contain data that might be considered confidential (e.g., data that is protected by legal privilege or by doctor-patient confidentiality, data that includes the content of individuals' non-public communications)?**
  No.

- **Does the dataset contain data that, if viewed directly, might be offensive, insulting, threatening, or might otherwise cause anxiety?**
  No.

- **Does the dataset relate to people?**
  Yes.

- **Does the dataset identify any subpopulations (e.g., by age, gender)?**
  No.

- **Does the dataset contain data that might be considered sensitive in any way (e.g., data that reveals race or ethnic origins, sexual orientations, religious beliefs, political opinions or union memberships, or locations; financial or health data; biometric or genetic data; forms of government identification, such as social security numbers; criminal history)?**
  No. The source datasets are already de-identified.

### E.3 Collection process

- **How was the data associated with each instance acquired?**
  We directly used the original data instance to curate our own dataset.

- **What mechanisms or procedures were used to collect the data (e.g., hardware apparatuses or sensors, manual human curation, software programs, software APIs)?**
  We mainly used Python scripts to collect, process and label the data.

- **If the dataset is a sample from a larger set, what was the sampling strategy (e.g., deterministic, probabilistic with specific sampling probabilities)?**
  The random sampling involved in this study relies on specific seed (42), thus becomes deterministic.

- **Who was involved in the data collection process (e.g., students, crowd workers, contractors), and how were they compensated (e.g., how much were crowd workers paid)?**
  The data collection process was fully performed by the study's authors.

- **Over what timeframe was the data collected?**
  N/A

- **Were any ethical review processes conducted (e.g., by an institutional review board)?**
  N/A.

- **Does the dataset relate to people?**
  Yes.

- **Did you collect the data from the individuals in question directly, or obtain it via third parties or other sources (e.g., websites)?**
  All data are collected through open-source database without interaction with individuals.

- **Were the individuals in question notified about the data collection?**
  N/A.

- **Did the individuals in question consent to the collection and use of their data?**
  N/A.

- **If consent was obtained, were the consenting individuals provided with a mechanism to revoke their consent in the future or for certain uses?**
  N/A.

- **Has an analysis of the potential impact of the dataset and its use on data subjects (e.g., a data protection impact analysis) been conducted?**
  The dataset does not have individual-specific information.

## E.4  Preprocessing/cleaning/labeling

- **Was any preprocessing/cleaning/labeling of the data done (e.g., discretization or bucketing, tokenization, part-of-speech tagging, SIFT feature extraction, removal of instances, processing of missing values)?**
  Yes. The preprocessing on MIMIC-III data follows existing work [20].

- **Was the "raw" data saved in addition to the preprocess/cleaned/labeled data (e.g., to support unanticipated future uses)?**
  N/A.

- **Is the software that was used to preprocess/clean/label the data available?**
  Preprocessing, cleaning, and labeling are done via Python.

## E.5  Uses

- **Has the dataset been used for any tasks already?**
  No.

- **Is there a repository that links to any or all papers or systems that use the dataset?**
  No.

- **What (other) tasks could the dataset be used for?**
  While the dataset is curated for research on federated medical knowledge injection problem, other studies concerning developing centralized medical foundation model can also leverage the dataset.

- **Is there anything about the composition of the dataset or the way it was collected and preprocessed/cleaned/labeled that might impact future uses?**
  N/A.

- **Are there tasks for which the dataset should not be used?**
  N/A.

## E.6  Distribution

- **Will the dataset be distributed to third parties outside of the entity (e.g., company, institution, organization) on behalf of which the dataset was created?**
  No.

- **How will the dataset be distributed?**
  The preprocessing code is available at `https://github.com/psudslab/FEDMEKI/tree/main/data_preprocess`. Users can download corresponding dataset and utilize the preprocessing scripts for generating the final dataset used in this study.

- **Will the dataset be distributed under a copyright or other intellectual property (IP) license, and/or under applicable terms of use (ToU)?**
  The dataset is released under MIT License.

- **Have any third parties imposed IP-based or other restrictions on the data associated with the instances?**
  No.

- **Do any export controls or other regulatory restrictions apply to the dataset or to individual instances?**
  No.

### E.7 Maintenance

- **Who will be supporting/hosting/maintaining the dataset?**
  The authors of this paper.

- **How can the owner/curator/manager of the dataset be contacted(e.g., email address)?**
  Contact the first authors (jqwang@psu.edu and xcwang@psu.edu).

- **Is there an erratum?**
  No.

- **Will the dataset be updated (e.g., to correct labeling erros, add new instances, delete instances)?**
  If any corrections are required, our plan is to upload an updated version of the dataset with comprehensive explanations for the changes. Furthermore, as we broaden our QA scope, we will consistently update the dataset with new QA templates/instances.

- **If the dataset relates to people, are there applicable limits on the retention of the data associated with the instances (e.g., were the individuals in question told that their data would be retained for a fixed period of time and then deleted)?**
  N/A

- **Will older versions of the dataset continue to be supported/hosted/maintained?**
  Primarily, we plan to maintain only the most recent version of the dataset. However, under certain circumstances, such as significant updates to our dataset or the need for validation of previous research work using older versions, we will exceptionally preserve previous versions of the dataset for up to one year.

- **If others want to extend/augment/build on/contribute to the dataset, is there a mechanism for them to do so?**
  Contact the authors of this paper.

# F  Training Task – Lung Opacity Detection

## F.1  Task Description

In the United States, pneumonia keeps the ailment on the list of top 10 causes of death in the country. The task is to locate lung opacities on chest radiographs. In this challenge [17], 18,406 images are annotated as either Lung Opacity or Normal, providing a basis for extracting the binary classification task. The task is to develop an algorithm to detect visual indicators of pneumonia in medical images. Specifically, the algorithm needs to identify and localize lung opacities in chest radiographs.

## F.2  License and Ethics

This dataset is permitted to access and utilize these de-identified imaging datasets and annotations for academic research, educational purposes, or other commercial or non-commercial uses, provided you adhere to the appropriate citations.

## F.3  Access and Preprocessing

The resource is available to access via the official website at `https://www.rsna.org/rsnai/ai-image-challenge/rsna-pneumonia-detection-challenge-2018` and Kaggle at `https://www.kaggle.com/c/rsna-pneumonia-detection-challenge/overview`. It includes dataset description, annotations, and mapping from RSNA image dataset to original NIH dataset. The data is organized as a set of patient IDs with corresponding image class annotations, including "No Lung Opacity/Not Normal," "Normal," and "Lung Opacity." We collected images labeled as either "Normal" or "Lung Opacity" and formulated the problem as a binary classification task. The code for preprocessing is available at `https://github.com/psudslab/FEDMEKI/tree/main/data_preprocess`.

## F.4  Data Samples

We provide a random data sample from the dataset and visualize it in Figure 2.

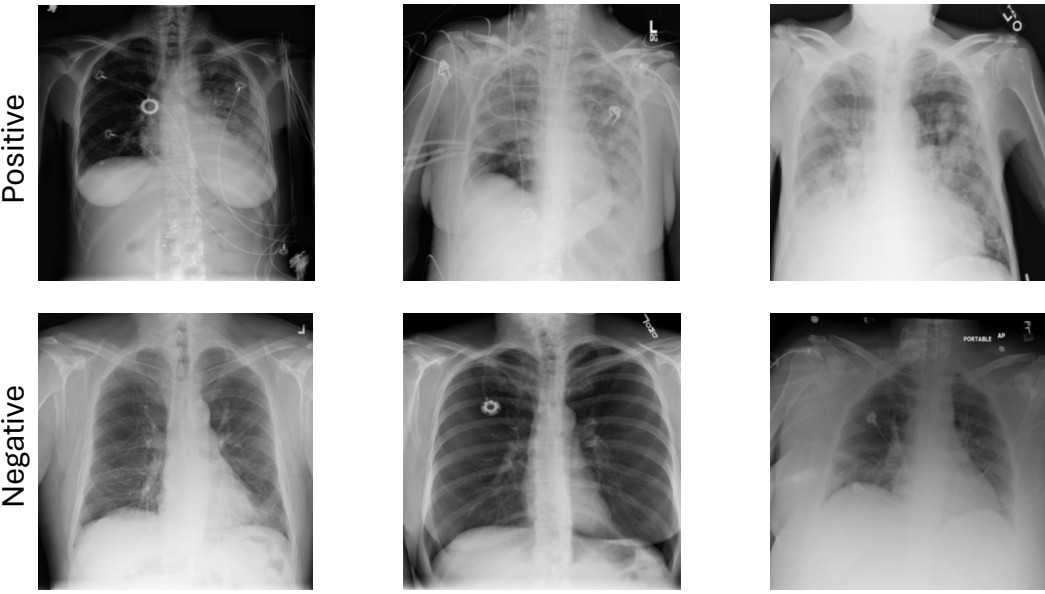

Figure 2: Data sample of lung opacity detection.

# G Training Task – COVID-19 Detection

## G.1 Task Description

This task challenges the model to assess whether X-ray images display symptoms of Covid-19, thereby evaluating the model's proficiency in interpreting medical imagery. For this purpose, we employ the COVQU dataset [18].

## G.2 License and Ethics

The licensing and ethical compliance adhere to the regulations established by the original datasets.

## G.3 Access and Preprocessing

This dataset can be accessed via the link at `https://www.kaggle.com/datasets/tawsifurrahman/covid19-radiography-database`. This dataset, featuring COVID-19, normal, and other lung infection categories, is being released incrementally. The initial release comprised 219 COVID-19, 1,341 normal, and 1,345 viral pneumonia chest X-ray (CXR) images. The first update expanded the COVID-19 category to include 1,200 CXR images. In the second update, the collection was further enlarged to include 3,616 COVID-19 positive cases, along with 10,192 normal, 6,012 lung opacity (non-COVID lung infection), and 1,345 viral pneumonia images, complete with corresponding lung masks. We selected normal and COVID-19 positive images to formulate this task as a binary classification problem.

## G.4 Data Samples

We provide a random data sample from the dataset and visualize it in Figure 3.

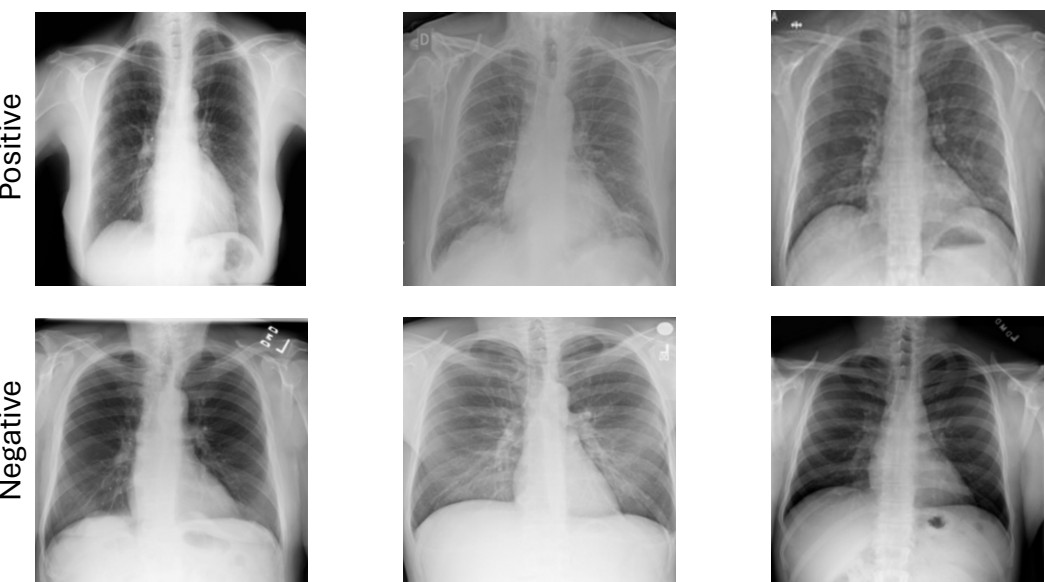

Figure 3: Data sample of Covid-19 detection.

# H    Training Task – ECG Abnormal Detection

## H.1    Task Description

Electrocardiography (ECG) is a crucial diagnostic tool for assessing a patient's cardiac condition, and automatic ECG interpretation algorithms offer significant support to medical personnel given the volume of ECGs routinely performed. The task involves analyzing the PTB-XL ECG dataset to develop and evaluate automatic ECG interpretation algorithms. We offer training and test set splits to facilitate algorithm comparability and include extensive metadata on demographics, infarction characteristics, diagnostic likelihoods, and signal properties, making it a comprehensive resource for training and evaluating automatic ECG interpretation algorithms.

## H.2    License and Ethics

The Institutional Ethics Committee approved the publication of the anonymous data in an open-access database (PTB-2020-1).

## H.3    Access and Preprocessing

The dataset can be directly downloaded with granted permission at `https://physionet.org/content/ptb-xl/1.0.3/` or via the terminal by `wget -r -N -c -np https://physionet.org/files/ptb-xl/1.0.3/`. Raw signal data was recorded in a proprietary compressed format, encompassing the standard set of 12 leads (I, II, III, AVL, AVR, AVF, V1, ..., V6) with reference electrodes on the right arm. Corresponding metadata, including age, sex, weight, and height, was systematically gathered in a database. Each ECG record includes a report, either generated by a cardiologist or automatically by the ECG device, which was then translated into a standardized set of SCP-ECG statements (scp_codes). For the relevant metadata, it is saved as one row per record identified by ecg_id. Totally, there are 28 columns categorized into identifiers, general metadata, ECG statements, signal metadata, and cross-validation folds. Additional details such as the heart's axis and stages of infarction (if applicable) were also documented. To ensure privacy and compliance with HIPAA standards, all personal information, including names of cardiologists and nurses, recording locations, and patient ages (with ages over 89 years reported within a 300-year range), was pseudonymized.

## H.4    Data Samples

We provide a random data sample from the dataset and visualize it in Figure 4. Here, a positive result indicates the presence of an abnormality in the ECG signal, while a negative result represents a normal signal. All signals are 12-channel, derived from the standard set of 12 leads (I, II, III, aVL, aVR, aVF, V1, ..., V6) with reference electrodes on the right arm.

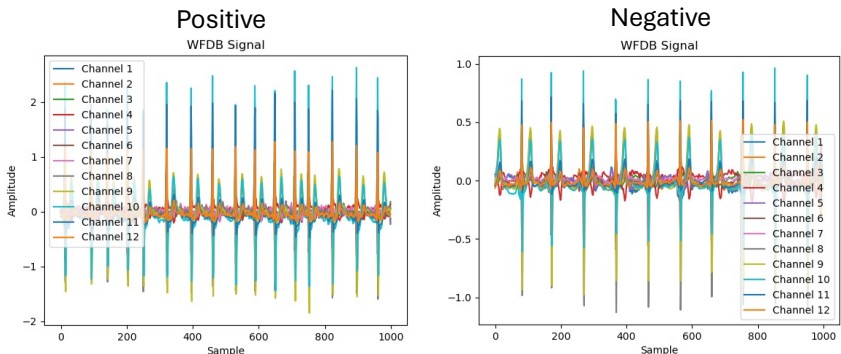

Figure 4: Data sample of ECG abnormal detection.

# I  Training Task – Mortality Prediction

## I.1  Task Description

The data source is MIMIC-III(Medical Information Mart for Intensive Care III), which is a substantial, anonymous, and publicly accessible repository of medical records. Each entry in the dataset contains ICD-9 codes that categorize the diagnoses and procedures conducted. In our work, we use the processed dataset to conduct the mortality prediction task.

## I.2  License and Ethics

The dataset is available for non-profit use in accordance with the license at `https://www.physionet.org/content/mimiciii/view-license/1.4/`.

## I.3  Access and Preprocessing

MIMIC-III can be accessed as a credentialed user on PhysioNet with an approved application at `https://mimic.mit.edu/`. In our experiment, we follow the ICU-oriented preprocessing pipeline [78] to process the data and follow the feature extraction pipeline [20] to extract dynamic features. Features extracted from this MIMIC-III database are listed in Table 5.

## I.4  Data Samples

We provide a random data sample from the dataset and visualize it in Figure 5.

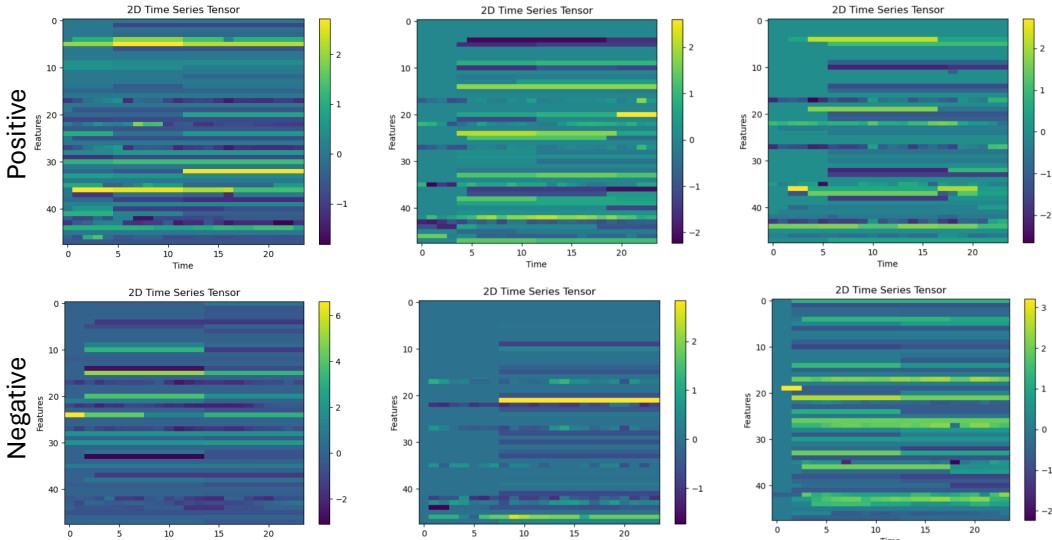

Figure 5: Data sample of mortality prediction.

Table 5: Clinical concepts extracted from MIMIC-III database [23]. The information is based on Table 15 provided in [20].

| Feature | RICU | Unit |
|---|---|---|
| Blood pressure (systolic) | sbp | mmHg |
| Blood pressure (diastolic) | dbp | mmHg |
| Heart rate | hr | beats/minute |
| Mean arterial pressure | map | mmHg |
| Oxygen saturation | o2sat | % |
| Respiratory rate | resp | breaths/minute |
| Temperature | temp | ∘C |
| Albumin | alb | g/dL |
| Alkaline phosphatase | alp | IU/L |
| Alanine aminotransferase | alt | IU/L |
| Aspartate aminotransferase | ast | IU/L |
| Base excess | be | mmol/L |
| Bicarbonate | bicar | mmol/L |
| Bilirubin (total) | bili | mg/dL |
| Bilirubin (direct) | bili_dir | mg/dL |
| Band form neutrophils | bnd | % |
| Blood urea nitrogen | bun | mg/dL |
| Calcium | ca | mg/dL |
| Calcium ionized | cai | mmol/L |
| Creatinine | crea | mg/dL |
| Creatinine kinase | ck | IU/L |
| Creatinine kinase MB | ckmb | ng/mL |
| Chloride | cl | mmol/L |
| CO2 partial pressure | pco2 | mmHg |
| C-reactive protein | crp | mg/L |
| Fibrinogen | fgn | mg/dL |
| Glucose | glu | mg/dL |
| Haemoglobin | hgb | g/dL |
| International normalised ratio (INR) | inr_pt | - |
| Lactate | lact | mmol/L |
| Lymphocytes | lymph | % |
| Mean cell haemoglobin | mch | pg |
| Mean corpuscular haemoglobin concentration | mchc | % |
| Mean corpuscular volume | mcv | fL |
| Methaemoglobin | methb | % |
| Magnesium | mg | mg/dL |
| Neutrophils | neut | % |
| O2 partial pressure | po2 | mmHg |
| Partial thromboplastin time | ptt | sec |
| pH of blood | ph | - |
| Phosphate | phos | mg/dL |
| Platelets | plt | 1,000 / μL |
| Potassium | k | mmol/L |
| Sodium | na | mmol/L |
| Troponin T | tnt | ng/mL |
| White blood cells | wbc | 1,000 / μL |
| Fraction of inspired oxygen | fio2 | % |
| Urine output | urine | mL |

# J  Validation Task – Enlarged Cardiomendiastinum Detection

## J.1  Task Description

This task is designed to evaluate the probability of an enlarged cardiomediastinum by using medical images from clinical assessments. It serves to gauge the model's ability to interpret radiographs effectively. The data for this task are sourced from the CheXpert Dataset [21]. CheXpert is a collection of 224,316 chest radiographs from 65,240 patients who underwent radiographic examinations at Stanford Health Care from October 2002 to July 2017. These images were gathered from both inpatient and outpatient centers and include the associated radiology reports.

## J.2  Access and Preprocessing

This dataset can be accessed via the link at `https://aimi.stanford.edu/chexpert-chest-x-rays` and downloaded via the link at `https://stanfordaimi.azurewebsites.net/datasets/8cbd9ed4-2eb9-4565-affc-111cf4f7ebe2`. The training set comprises 224,316 high-quality X-ray images from 65,240 patients, annotated to automatically identify 14 different observations from radiology reports, reflecting the inherent uncertainties of radiographic interpretation. The validation set includes 234 images from 200 patients, each manually annotated by three board-certified radiologists. The test set, which remains unreleased to the public and is held by the organizers for final assessment, contains images from 500 patients annotated through the consensus of five board-certified radiologists. CheXpert images have an average resolution of 2828x2320 pixels.

## J.3  Data Samples

We provide a random data sample from the dataset and visualize it in Figure 6.

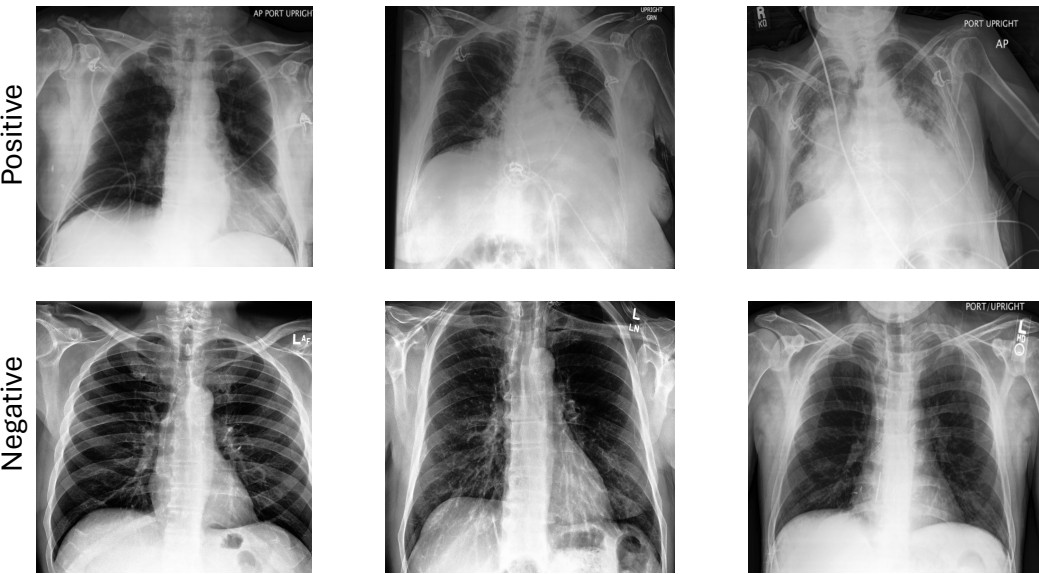

Figure 6: Data sample of enlarged cardiomendiastinum detection.

# K    Validation Task – Sepsis Prediction

## K.1    Task Description

This task focuses on predicting the likelihood of sepsis during ICU stays, assessing the model's ability to analyze various clinical data, including lab events, diagnoses, and prescriptions. For this research, we utilize the MIMIC-III database, extracting features and cohorts through a well-established preprocessing pipeline [20].

## K.2    License and Ethics

This dataset is governed by the license available at the following URL: `https://www.physionet.org/content/mimiciii/view-license/1.4/`.

## K.3    Access and Preprocessing

MIMIC-III dataset can be accessed with the approved permission via `https://mimic.mit.edu/`. A random sampling strategy is applied to select a subset with 1,000 samples for testing.

## K.4    Data Samples

We provide a random data sample from the dataset and visualize it in Figure 7. It has clinical features only, including lab events and vital signs. Although the data feature space of both mortality prediction and sepsis prediction is the same, the feature distributions are significantly different. That is why the zero-shot inference on this task performs worse than the mortality prediction, as shown in Table 4.

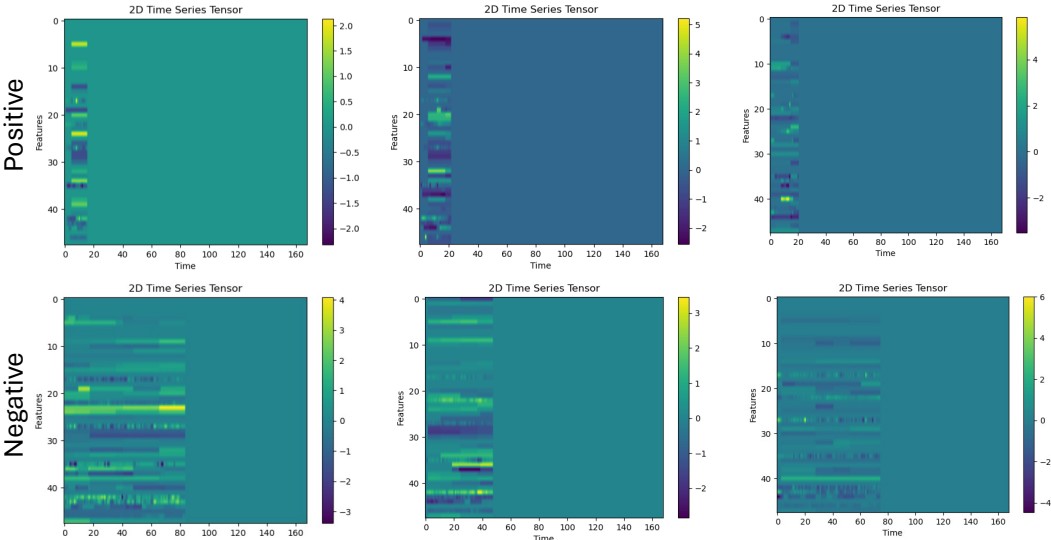

Figure 7: Data sample of sepsis prediction.

# L  Validation Task – MedVQA

## L.1  Task Description

The SLAKE dataset is designed for Medical Visual Question Answering (Med-VQA), integrating detailed visual and textual annotations with a medical knowledge base. It features semantic segmentation masks and object detection bounding boxes for each radiology image. SLAKE includes both basic clinical and complex compositional questions, and is uniquely bilingual in English and Chinese. It expands coverage to more body parts and introduces new question types related to shape and knowledge graphs, with comparative data provided against the VQA-RAD dataset. In this task, the model uses both visual context and verbal questions as inputs, requiring answers that integrate textual questions and visual context. This task tests the model's ability to align text and image modalities in the medical domain.

## L.2  License and Ethics

Ethical approval was not required as confirmed by the license attached with the open access data in [12].

## L.3  Access and Preprocessing

This dataset can be accessed at `https://www.med-vqa.com/slake/`. For the image part, 642 images, including 12 diseases and 39 organs, were in the format of CTs and MRIs. With the help of a constructed knowledge graph, it covers questions with ten different content types and semantic labels proposed by doctors. We randomly select 1000 samples from the test dataset.

## L.4  Data Samples

We provide a random data sample from the dataset. Question-answering pair and corresponding image (Figure 9) are listed below. This MedVQA dataset contains different types of images except for chest X-ray images, which are different from the ones we used in the model training. In addition, the trained FEDMEKI does not use any medical question-answering training tasks. Therefore, its performance of this "new" task is limited, as shown in Table 4.

Q: *How can you tell this is a T2 weighted image?*
A: *CSF is white.*

Q: *Is the heart enlarged?*
A: *No.*

Q: *What it causing the widening?*
A: *Mass*

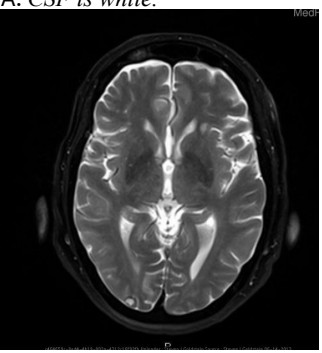
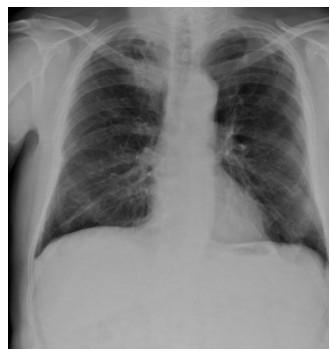
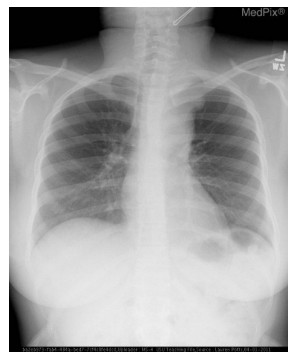

Figure 8: Data sample of MedVQA task.

# M   Validation Task – Signal Noise Clarification

## M.1   Task Description

This task is dedicated to precisely characterizing noise in ECG signals through a question-and-answer format.

## M.2   License and Ethics

The Institutional Ethics Committee approved the publication of the anonymous data in an open-access database (PTB-2020-1).

## M.3   Access and Preprocessing

It utilizes data from an established ECG question answering dataset [22] and a related ECG database [19], which can be accessed via `https://physionet.org/content/ptb-xl/1.0.3/` and `https://github.com/Jwoo5/ecg-qa/tree/master/ecgqa/ptbxl`. The ECG signals used in this task consist of 12 channels and have a duration of 10 seconds, mirroring the parameters used in the ECG Abnormal Detection task. We randomly sample 1,000 ECG-question pairs as the validation data.

## M.4   Data Samples

We provide a random data sample from the dataset. Question-answering pair and corresponding signal (Figure 9) are listed below. Although we have a training task on ECG, the ECG abnormal detection task is different from this one. This task aims to answer the noise types of ECG signals according to the input ECG and the question. We can see that the ECG signals in Figure 9 are quite different from the ones in Figure 4, which increases the difficulty of this task significantly.

Q: *What kind of noises does this ECG show in lead aVL?*
A: *baseline drift, static noise.*

Q: *Does this ECG show baseline drift?*
A: *Yes.*

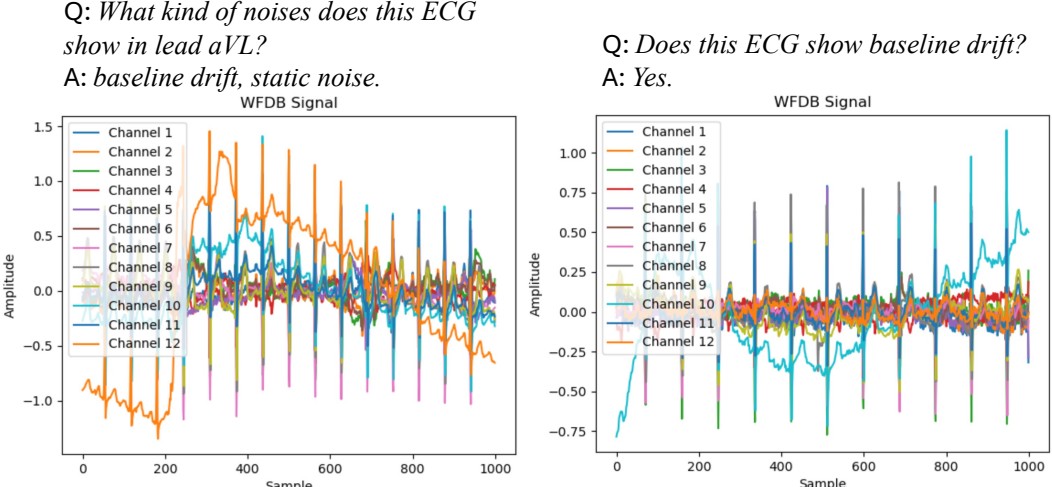

Figure 9: Data sample of signal noise clarification.

# N Implementation Details

## N.1 Model Details

For each local model $\mathbf{W}_n$ deployed in client $C_n$, we implement their modality-specific encoders and task-specific decoders. Details about encoders for different modalities can be found in Table N. As all training tasks can be categorized as binary classification, we use MLPs as their task-specific decoders, where decoders for different tasks do not share any parameter.

Table 6: Details of modality-specific encoders.

| Modality | Encoder | # of Parameters |
|---|---|---|
| Image | Deit-tiny [79] | 7.8M |
| Signal | CNN [80] | 4.1M |
| Vital Sign | Transformer [81] | 3.7M |
| Lab Results | Transformer [81] | 3.7M |
| Input | Transformer [81] | 3.7M |
| Output | Transformer [81] | 3.7M |

## N.2 Optimizer Hyperparameters

We leverage Adam optimizer [82] for optimizing both local model $\mathbf{W}_n$ and foundation model $\mathcal{F}$. The number of communication rounds is set to 10. For local model $\mathbf{W}_n$, we find the learning rate of 1e-4 for local models achieves a decent convergence, while the learning rate for the foundation model is configured to 5e-4. The batch size of both the foundation model and local models is set to 64.

## N.3 Task Prompts

Task prompts for classification tasks are listed in Table 7. For MedVQA and signal noise clarification, prompts are questions themselves.

Table 7: Task Prompts.

| Task Name | Prompt |
|---|---|
| Lung Opacity Detection | Assess this CT image: should it be classified as lung opacity? |
| Covid-19 Detection | Based on this image, is the patient COVID-19 positive? |
| ECG Abnormal Detection | Is the given ECG abnormal? |
| Mortality Prediction | Based on these clinical features, will mortality occur in this patient? |
| Enlarged Cardiomendiastinum Detection | Does this image show evidence of enlarged cardiomediastinum? |
| Sepsis Prediction | Based on these clinical features, will sepsis occur in this patient? |

## N.4 Baselines

To better understand the benchmarks used in the experiments, we use visualizations to demonstrate each approach clearly.

For single-task evaluation, we use FedAvg$_s$/FedProx$_s$ (Figure 10), FedAvg$_s^+$/FedProx$_s^+$ (Figure 11), FedAvg$_s^*$/FedProx$_s^*$, and FedAvg$_s^{\mathcal{F}}$/FedProx$_s^{\mathcal{F}}$ (Figure 12). When training single tasks, we only use each task data as the model input. For multi-task training, we also have eight baselines that are shown from Figure 13 to Figure 15. These models will train all the task data together.

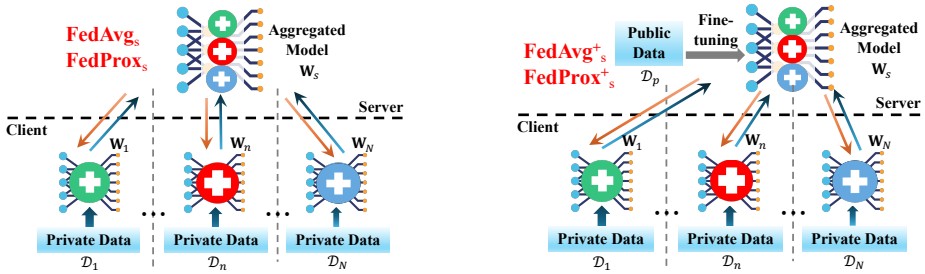

Figure 10: **FedAvg**$_s$ or **FedProx**$_s$     Figure 11: **FedAvg**$_s^+$ and **FedProx**$_s^+$

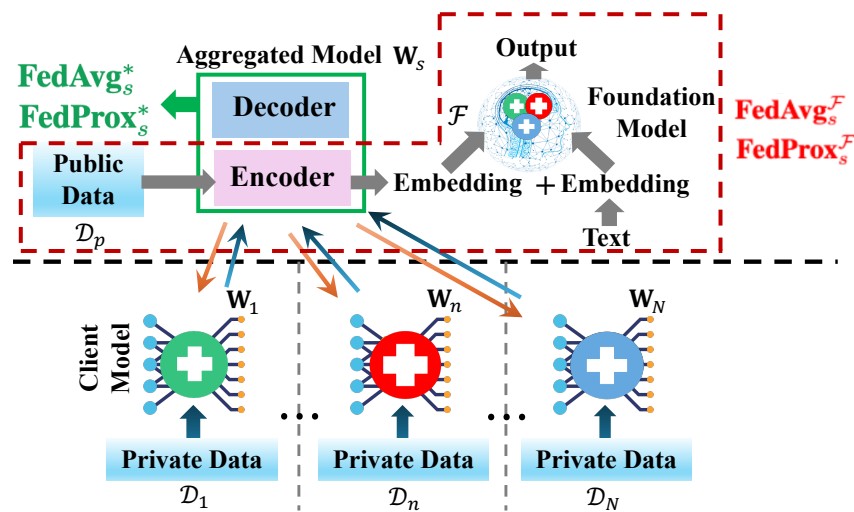

Figure 12: **FedAvg**$_s^{\mathcal{F}}$/**FedProx**$_s^{\mathcal{F}}$ (red dot line) and **FedAvg**$_s^*$/**FedProx**$_s^*$ (green line).

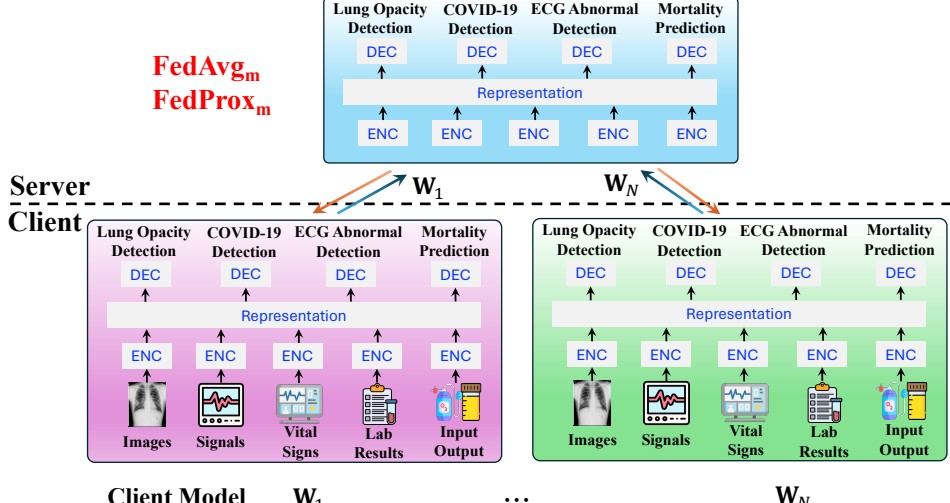

Figure 13: **FedAvg**$_m$ or **FedProx**$_m$

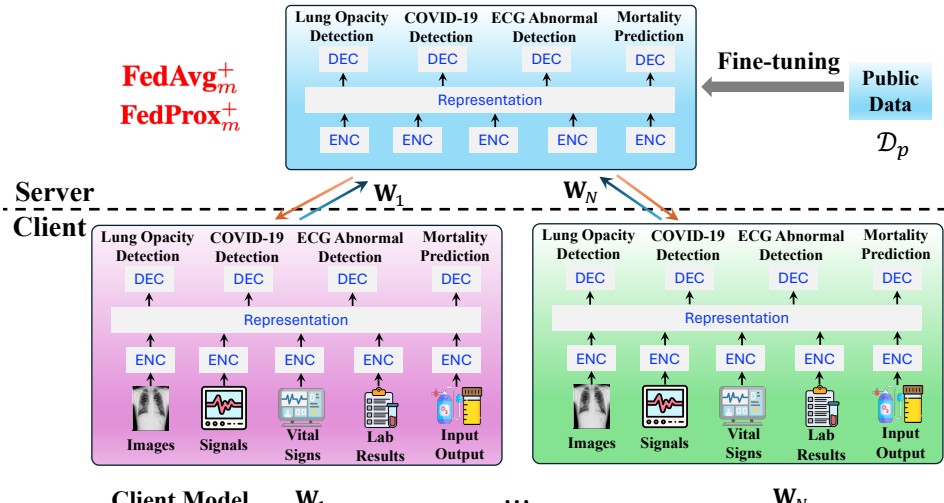

Figure 14: **FedAvg**$_m^+$ and **FedProx**$_m^+$

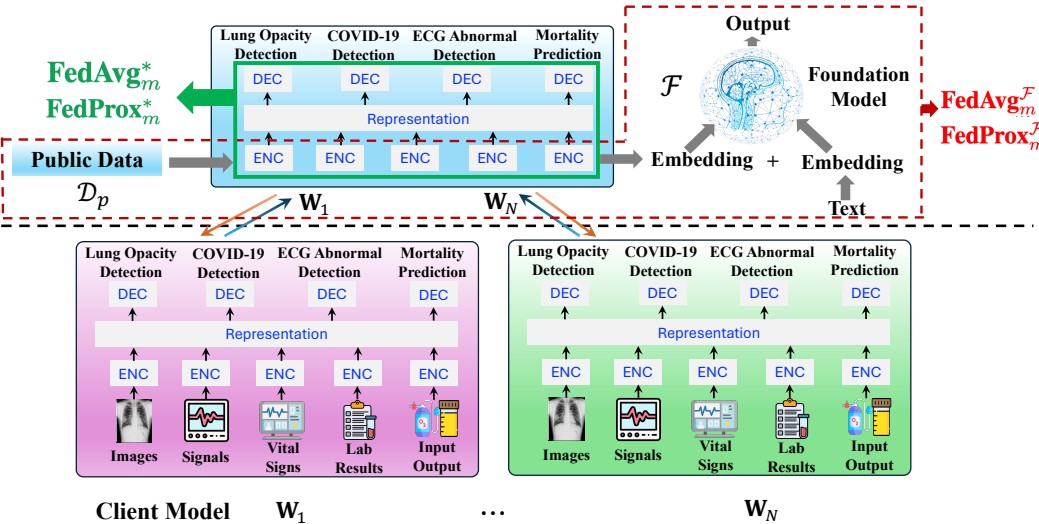

Figure 15: **FedAvg**$_m^{\mathcal{F}}$/**FedProx**$_m^{\mathcal{F}}$ (red dot line) and **FedAvg**$_m^*$/**FedProx**$_m^*$ (green line).

