# OpenReview forum: "FEDMEKI: A Benchmark for Scaling Medical Foundation Models via Federated Knowledge Injection"
_NeurIPS.cc/2024/Datasets_and_Benchmarks_Track — NeurIPS 2024 Track Datasets and Benchmarks Spotlight_

### Official Review · Reviewer_9Bk2 · 2024-07-06
**Review of Submission1236**

**Rating:** 8
**Confidence:** 5

**Review:**

The writing quality of the paper is very high, with reasonable and correct statements. It explores for the first time the scalability of medical foundation models within a federated framework, laying a foundation for continued research and development in this field.
Pros:
- Privacy Preservation: The use of federated learning ensures that sensitive medical data remains decentralized, addressing privacy concerns effectively.
- Comprehensive Benchmarking: The paper provides a thorough evaluation of the FEDMEKI platform using multiple benchmark approaches, demonstrating its utility and effectiveness.
- Multi-modality and Multi-task Coverage: The inclusion of various medical modalities and tasks in the dataset ensures the robustness and versatility of the FEDMEKI platform.
- Scalability: The platform's design allows for scalability, enabling the integration of new medical tasks and modalities as needed.
- Writing: This paper is well-structured. It is easy to follow.

Cons:
- Performance Trade-off: There is a noticeable performance trade-off when injecting medical knowledge into foundation models, with federated scaled models performing worse than traditional federated learning approaches on some tasks.
- Zero-shot Evaluation: The performance on zero-shot inference tasks remains unsatisfactory, indicating room for improvement in handling unseen tasks and modalities.
- The overhead of communication and synchronization is high.

**Strengths:**

Refer to the Review above.

**Additional Feedback:**

- Optimization of Training Algorithms: Focus on improving the efficiency and performance of the training algorithms to handle the complexities of federated learning better.
- Efficient Communication Protocols: Develop more efficient communication protocols to reduce overhead and improve synchronization across clients.

**Clarity:**

The writing quality is high, with clear explanations of concepts, methodologies, and results. The paper is well-structured, making it easy to follow the logical progression of the research. The inclusion of figures and tables aids in the understanding of the FEDMEKI platform and its evaluation.

**Correctness:**

The statements in the paper appear accurate, with the methodologies and results being well-supported by experimental data. The construction of the dataset is reasonable, leveraging publicly available sources and ensuring a diverse representation of medical tasks and modalities. The evaluation methods and experimental designs are appropriate, with comprehensive benchmarks provided for both training and validation tasks.

**Documentation:**

The dataset is detailed in terms of data collection, with clear descriptions of the sources and the types of medical data included. The organization of the dataset for training and validation tasks is well-documented, ensuring usability and maintenance. Furthermore, this paper provides sufficient details to support reproducibility.

**Ethics:**

No.

**Limitations:**

The authors adequately discuss the limitations of their work, including the performance trade-offs and challenges related to data heterogeneity and communication overhead in federated learning. They also mention the computational efficiency and scalability issues.

**Opportunities For Improvement:**

Refer to the Review above.

**Relation To Prior Work:**

The paper clearly differentiates its work from previous contributions by focusing on the novel task of federated medical knowledge injection into foundation models. The emphasis on privacy-preserving methods and the integration of diverse medical modalities and tasks sets this work apart from existing medical foundation models.

**Summary And Contributions:**

This paper presents a new benchmark to address the challenge of integrating medical knowledge into foundation models under privacy constraints. The design and implementation of this benchmark are comprehensive, capable of handling multi-site, multi-modal, and multi-task medical data. By leveraging a broader spectrum of medical knowledge while preserving data privacy, this benchmark significantly enhances the capabilities of medical foundation models, demonstrating wide-ranging application prospects. Specifically, the contributions of this paper are as follows:
- Curated Comprehensive Dataset: The authors curate a dataset from seven publicly available medical sources, covering eight diverse medical tasks across seven modalities.
- Benchmarking and Validation: The paper implements 16 benchmark approaches to validate the FEDMEKI platform, including traditional federated learning, federated learning with fine-tuning, and federated learning with foundation model scaling.
- Enhancement of Medical Foundation Models: The FEDMEKI platform is shown to enhance the capability of medical foundation models by allowing them to learn from a broader spectrum of medical knowledge without direct data exposure.

---

> ### Author Rebuttal · Authors · 2024-08-16
>
> We do appreciate the constructive suggestions. We would like to provide our replies one by one below.
>
> `>>> C1`
>
> Firstly, as discussed in Section 5.2.1, the lower performance of federated scaled models is reasonable. Traditional approaches are specifically designed for federated learning, resulting in homogeneous aggregated models without any noise introduced during knowledge injection. In contrast, the knowledge injection between the aggregated model and the foundation model involves significantly heterogeneous structures. This explains why the performance of naive knowledge injection navigation cannot compete with mature federated learning approaches.
>
> Secondly, as shown in Table 4, traditional federated learning approaches cannot handle unseen tasks, whereas the knowledge injection approach can. This observation indicates that the knowledge injection approach possesses indispensable characteristics lacking in conventional FL approaches. Although temporarily weaker, it is a more comprehensive method for tackling complex medical scenarios, where numerous tasks and the ability to address unencountered tasks are crucial.
>
> Thirdly, as stated in Section 5.2.1, achieving optimal performance in one step is not the objective of this study. Our goal is to develop the knowledge injection platform for future exploration. We will provide a more detailed explanation in the camera-ready version of this manuscript.
>
>
> `>>> C2`
>
> Results from the zero-shot evaluation indicate that the foundation model becomes capable of solving unseen tasks after the knowledge injection. Achieving extremely high results for knowledge injection is not the primary objective of this work. In alignment with the spirit of this track, our aim is to provide a platform for colleagues to develop more advanced knowledge injection approaches. The proposed knowledge injection methods serve as an initial solution rather than our major contribution. We will provide further clarification in Sections 4 and 5 accordingly.
>
> `>>> C3`
>
> Thank you for your valuable feedback. In the current version of our study, we concentrate on the cross-silo setting, where both the number of clients and communication rounds are relatively limited. Furthermore, compared with directly exchanging large foundation models between the server and the clients, our approach only needs to exchange the local compact models to save the communication cost. In future work, we plan to explore strategies to further reduce communication costs and to consider incorporating asynchronous settings as the reviewer suggested.
>
> `>>> Limitations`
>
> Thank you for your insightful comments regarding the limitations of our work. We are committed to further exploring this direction and will endeavor to address these limitations in our future research.

---

> > ### Comment · Reviewer_9Bk2 · 2024-08-30
> >
> > Thanks to the authors for their detailed and thorough responses to my comments and questions. I believe this addresses my core concerns. I think this paper is well-structured, has clear motivations, thorough experiments, and offers new insights into the development of the field.

---

### Official Review · Reviewer_XgYN · 2024-07-16
**Review of Paper #1236**

**Rating:** 7
**Confidence:** 5
**Correctness:** The claims made in the paper are corr…
**Clarity:** Yes, the paper is well written.

**Review:**

The comments are as follows:
The platform addresses the challenge of dealing with sensitive medical data, which is difficult to gather in large quantities needed for training robust medical foundation models. FedMEKI offers a convincing solution to this issue.
Experiments conducted using the platform on different medical datasets show its capability to handle multiple tasks effectively.
The paper is well-presented, with clear diagrams, figures, tables, and writing.
There is curiosity about whether other federated learning approaches could be integrated into this platform.

**Strengths:**

1. The research topic is novel and intriguing, holding significant theoretical and practical implications.
2. The findings are robustly supported through detailed experiments, thorough analysis, and discussions.
3. The paper encompasses a wide range of tasks, data modalities, and federated learning methodologies.
4. FedMEKI is well-documented and user-friendly, enhancing its accessibility and usability.

**Additional Feedback:**

N/A

**Documentation:**

This is the benchmark paper. They provide sufficient details to help with reproducibility.

**Limitations:**

See the opportunities for improvement.

**Opportunities For Improvement:**

The results of MedVQA and Signal Noise Clarification in Table 4 are much lower than those of the first medical image task. The author should provide more explanations.
The authors need to explain why the sepsis task's performance is lower than that of the mortality prediction task, even though they use the same feature space.
Typos: Table N -> Table 6 (Line 1353)

**Relation To Prior Work:**

Yes, the difference is clearly discussed.

**Summary And Contributions:**

The paper introduces FedMEKI, a federated learning-based platform designed specifically for medical-related tasks. FedMEKI aims to enhance the development of medical foundation models through collaborative efforts among various clients. The platform has been tested across multiple datasets and tasks to demonstrate its efficacy.

---

> ### Author Rebuttal · Authors · 2024-08-16
>
> `>>> Opportunities For Improvement`
>
> Thanks for the valuable comments. We divide tasks into two sets: the training set and the validation set. As emphasized in Section 4.3, the training tasks serve two purposes: injecting knowledge into the foundation model and providing evaluation for the scaled model. This means that the foundation model has already encountered these tasks, explaining the relatively high performance in Tables 2 and 3, where the model is evaluated in a fine-tuning manner. In contrast, the validation tasks, as referred to in Section 4.3, are solely for zero-shot evaluation, meaning the model has not encountered the tasks listed in Table 4. This explains the significantly lower performance observed in Table 4 compared to Tables 2 and 3. We will provide a more explicit introduction to our experimental setup, along with a more detailed analysis of the results.

---

### Official Review · Reviewer_cYFo · 2024-07-16
**Good benchmark for the field of medical foundation models.**

**Rating:** 7
**Confidence:** 4
**Correctness:** The evaluation methods and experiment…
**Clarity:** Yes. The paper is well-organized and …

**Review:**

The comments are as follows:
1. This work introduces a novel platform that allows medical data holders to collaborate on a medical foundation model without sharing the underlying data. The research question addressed is both timely and critical.
2. The platform's design is thoroughly detailed and systematically presented, showing potential for generalization across various federated learning scenarios and other contexts involving sensitive data.
3. Extensive experiments validate the platform's effectiveness. The authors also provide insightful discussions and analyses related to their findings.
4. It is recommended that the authors explore potential extensions of this platform to cross-device settings, which could include numerous wearable medical devices.

**Strengths:**

Strengths
1. The research question is both well-motivated and emergent, representing a significant area of interest within the community.
2. The application of federated learning in medical tasks is demonstrated as a promising approach.
3. The paper is clearly written, making it easy for readers to understand and follow the presented concepts.
4. Comprehensive documentation and the availability of a GitHub repository aid other researchers and engineers in utilizing and contributing to the project.

**Additional Feedback:**

N/A

**Documentation:**

The authors provide detailed documentation and Github.

**Limitations:**

This work has no potential negative societal impact. Please see the Opportunities for Improvement.

**Opportunities For Improvement:**

Please continue to update and maintain the GitHub repository as new medical data and tasks become available. Additionally, it’s highly recommended to consider enabling permissions to allow other users to contribute to the repository, fostering a collaborative and dynamic development environment.

**Relation To Prior Work:**

Yes. This work covers appropriate related works.

**Summary And Contributions:**

This paper introduces FedMEKI, a pioneering platform that addresses a novel yet practical research challenge. FedMEKI facilitates the integration of medical knowledge into foundation models while preserving privacy using federated learning techniques. The platform comprehensively addresses medical research by considering multi-site, multi-modal, and multi-task dimensions. The authors have conducted extensive experiments and provided detailed documentation to aid further research.

---

> ### Author Rebuttal · Authors · 2024-08-16
>
> `>>> Opportunities For Improvement`
>
> Thanks for the valuable feedback. We will continue maintaining the repository and release the permission to other people to contribute appropriately.
>
> `>>> Limitations`
>
> We will work on updating the GitHub repository and ensure the updates will not have any negative societal impact.

---

### Official Review · Reviewer_jMQ7 · 2024-07-17
**A Benchmark for Scaling Medical Foundation Models via Federated Knowledge Injection**

**Rating:** 8
**Confidence:** 4
**Clarity:** The paper is well written and easy to…

**Review:**

First, I believe the new task proposed in this paper is meaningful, especially in the medical field where sensitivity and privacy issues are paramount, which have not been considered by previous medical foundation models.

Secondly, the FEDMEKI platform proposed is also reasonably designed, with a well-constructed dataset and a sufficient number of baseline models.

Overall, the quality of the paper is good, with clear and fluent writing that allows readers to quickly grasp the research content and core contributions.

**Strengths:**

- The new task introduced is highly relevant, particularly in the medical field where addressing sensitivity and privacy issues is crucial.
- The FEDMEKI platform offers a novel and practical solution by federating medical knowledge injection into foundation models.
- The creation of a multi-site, multi-modal, multi-task dataset is a significant contribution, providing a robust benchmark for the new task.
- Thorough Baseline Implementation: Implementing 16 different approaches as benchmark baselines ensures that the platform is well-validated and facilitates reproducibility.

**Additional Feedback:**

None

**Correctness:**

All aspects of dataset construction, evaluation methods, and experiment design are correct and well-executed.

**Documentation:**

The paper provides sufficient detail on data collection and access, using open-source datasets. Additionally, it includes a GitHub project link, ensuring availability and supporting reproducibility.

**Limitations:**

The authors have appropriately discussed some of the potential limitations of their work in the Discussion section. I believe these are also areas worth improving in future research.

**Opportunities For Improvement:**

I have some questions regarding the construction of the dataset. Specifically, what are the reasons for the division of training tasks and validation tasks? Were they divided based on the difficulty levels of different tasks or considering specific attributes of the tasks? Or were the tasks divided randomly?

Since the construction of the dataset is the foundation for subsequent model performance evaluation, the same model may perform differently under different divisions of training tasks and validation tasks. Therefore, how does the proposed dataset division in this paper ensure practical and realistic applicability?

**Relation To Prior Work:**

Yes, the paper clearly discusses how this work differs from previous contributions in the Introduction and Related Work sections.

**Summary And Contributions:**

Traditional medical foundation models, despite achieving superior performance on various domain-specific tasks, often require fine-tuning existing general domain foundation models in a centralized training manner, which does not consider the sensitivity and privacy issues of medical data. Therefore, a more practical and realistic solution is to collaboratively inject medical knowledge learned from private client data into foundation models in a federated manner.

This paper introduces a new task to scale existing medical foundation models, named Federated Medical Knowledge Injection into foundation models.

To address the challenges and benchmark this new task, the authors curated a new multi-site, multi-modal, multi-task dataset. Additionally, they developed an open-source federated medical knowledge injection platform, FEDMEKI, to benchmark this new task with the curated dataset. They implemented 16 different approaches as benchmark baselines to validate the effectiveness of the FEDMEKI platform.

---

> ### Author Rebuttal · Authors · 2024-08-16
>
> We thank the reviewer for the valuable feedbacks.
>
> `>>> Opportunities For Improvement`
>
> We do not categorize datasets by difficulty levels; instead, we split them based on modalities. In the knowledge injection task, the medical foundation model gains from clients through parameters in modality-specific encoders. Ensuring that training tasks encompass all modalities allows the foundation model to absorb knowledge from all possible sources. Likewise, the model must be evaluated with all possible modalities to ensure a thorough assessment of knowledge injection via parameters in each modality-specific encoder.
>
> Consequently, we carefully divide the training and validation tasks, ensuring that the model has access to all medical modalities during the knowledge injection process and is evaluated with tasks that share these encountered modalities. This division provides a comprehensive evaluation of the medical knowledge injection task. We will provide a more detailed explanation on the rationale in Section 4.
>
> `>>> Limitations`
>
> Thanks for the feedback. We will improve and extend our work following what we discussed in the discussion section.

---

### Decision · Program_Chairs · 2024-09-26

**Decision:**

Accept (Spotlight)

**Comment:**

This paper initially received positive comments from all the reviewers. This paper provides a benchmark for medical-related tasks, which facilitates the integration of medical knowledge into foundation models with federated learning techniques. The paper is well-written, the research topic is novel and intriguing, and the platform FedMEKI is well-documented and user-friendly. Therefore, the AC agrees with the reviews and recommends acceptance.